# The Relationship between Parent-Offspring Communication and the School Adaptation of Leftover Children in Overseas Countries: The Mediating Role of Companionship and the Moderating Role of a Sense of Safety

**DOI:** 10.3390/bs13070557

**Published:** 2023-07-05

**Authors:** Huilan Zhang, Bingwei Shen, Chunkao Deng, Xiaojun LYu

**Affiliations:** College of Education, Wenzhou University, Wenzhou 325035, China; 21460410022@stu.wzu.edu.cn (H.Z.); 20210411121@stu.wzu.edu.cn (B.S.)

**Keywords:** parent-offspring communication, school adaptation, leftover children, companionship

## Abstract

Background: In the diasporic eastern coastal region of China, leftover children are a unique group of children; their social adaptation challenges are more prominent due to transnational separation from parents. This study explores the relationship between parent-offspring communication and school adaptation among leftover children. Methods: We administered questionnaires to 957 children from six schools in June and December of 2022. All students in the sample were randomly selected from within the classrooms. In total, 561 (47.95% female, mean age = 12.84, SD = 0.95) of them were leftover children. Self-report questionnaires on communication with their parents, school adaptation, companionship, and feelings of safety were used in this investigation We subsequently used SPSS software and the PROCESS plugin to analyze the relationships between variables. Results: A significant and positive relationship was found between parent-offspring communication and school adaptation in leftover children. Companionship mediated this effect. Additionally, the impact of parent-offspring communication on companionship was moderated by a sense of safety. Conclusions: The study concluded that parent-offspring communication, school adaptation, companionship, and a sense of safety were all positively correlated. In addition, companionship partially mediated the relationship between parent-offspring communication and school adaptation. Moreover, a sense of safety played a moderating role. These conclusions can provide empirical support for improving the school adaptation of leftover children.

## 1. Introduction

Overseas Chinese leftover children are a distinct group of leftover children in eastern coastal China, whose adaptation in school is not optimal due to the temporal and spatial separation from parents abroad and the educational conflicts that manifest under a transnational upbringing [1]. School adaptation is a behavioral manifestation of students’ feelings about school, their level of involvement, their academic achievement, and peer interactions in school. School adaptation usually includes three aspects: academic adaptation, psycho-behavioral adaptation, and social-emotional adaptation [2,3]. It was found that the ‘left behind’ background label lowered the students’ academic performance and school adaptation levels [4], and that negative factors such as behavioral deviations and psychological issues were significantly higher among leftover children than children living with their parents [5]. It was noted by Zhang et al. that transnational experiences of being left-behind led to more difficulties in adapting to school among leftover children in the Chinese diaspora [6]. For students, proper school adaptation facilitates the development of social competencies such as successful academic completion, maintenance of companionship, and acquisition of social values [7], while poor school adaptation leads to academic failure, behavioral issues, and social maladaptation [8]. In addition, in our fieldwork, we also found that children who were left-behind in overseas Chinese countries have a poorer level of school adaptation compared to children who were not left behind. Furthermore, the differences were statistically significant (*t* = −5.64, Cohen’s *d* = −0.73, *p* < 0.001) [9]. Consequently, an investigation of school adaptation’s influencing factors and mechanisms is necessary, and it is essential to improve the school adaptation levels of leftover children in overseas countries.

### 1.1. The Relationship between Parent-Offspring Communication and School Adaptation

Family socialization is the initial, most significant, and far-reaching aspect of the socialization of children and is extremely important for children’s psychological maturation, character formation, and acquisition of behavioral norms [10]. Family interactions are essential for children’s social interactions and reflect the effects of family socialization [11]. Healthy parent-offspring communication not only helps to provide for children’s need for love and belonging [12], but it also has an immeasurably important impact on the psychological health and character improvement of children, which can promote children’s conscious acceptance of their parents’ educational requirements and expectations and enable them to automatically internalize their parents’ education as their own internal motivation to achieve their own positive development [13]. In contrast, poor parent-offspring communication may limit the individual’s ability to regulate negative situations and inhibit the child from adopting a more adaptive approach to negative situations [14].

Life course theory refers to the principle of “interconnected lives”, in which each generation’s life situation is greatly influenced by others in its network of relationships [15]. Children are influenced by their parents in the process of adapting to their school and social environments, and effective parent-offspring communication can improve children’s comprehension of their parents and promote a tendency to adopt positive emotions when dealing with challenges, resulting in better adaptation to school. For leftover children in the Chinese diaspora, the “adverse cumulative effect” of reduced parent-offspring communication due to parental absence may lead to children’s difficulties in dealing with conflicts, poor interpersonal relationships, and poor academic performance, resulting in school maladaptation [16]. Previous studies have found that there is a strong relationship between parent-offspring communication and school adaptation of children in terms of social adaptation [17], academic achievement [18], and psychological adaptation [19], as well as within the group of leftover children [20]. Therefore, to investigate the relationship between parent-offspring communication and the academic adaptation of leftover children in the Chinese diaspora, Hypothesis 1 was put forward in this study.

**Hypothesis** **1 (H1).**
*The school adaptation of leftover children living in the Chinese diaspora is positively influenced by parent-offspring communication.*


### 1.2. The Mediating Role of Companionship

Companionship is an interpersonal relationship established by individuals of comparable age or psychological developmental level in the process of joint activities and collaboration [21], and they are an important environment in the process of individual interaction. Studies have found that the family system is the first environmental system that individuals come into contact with from birth, and it provides a reference for their socialization process in addition to meeting their material needs [22]. As a “micro-society” in children’s development, peers are an important influence in the socialization process alongside the family environment and school environment, and are largely influenced by the family environment [23]. An important aspect of the family environment is parent-offspring communication, which plays a key role in children’s development. Studies have found that parent-offspring communication positively predicted adolescents’ companionship and that children imitate their parents’ behaviors [24]. Healthy parental communication styles and content have a positive effect on children’s maintenance of companionship. In addition, on the basis of the spillover hypothesis, the parent-offspring communication of leftover children in the Chinese diaspora tends to be more “unfamiliar”, and the needs of family functions cannot be met; therefore, their family system partially spills over to the school system [25], leading to an increase in their peer group interactions, and parent-offspring communication is correlated with children’s companionship.

Past research has found a strong association between companionship and children’s academic achievement and social behavior [26,27]. High levels of academic adaptation and better academic performance are demonstrated by children who are welcomed by their peers; children with high levels of peer acceptance tend to show more pro-social behaviors and better behavioral adaptation. For leftover children in the Chinese diaspora, the absence of parenting leads children to be more willing to seek help from peers, and the importance of the role of companionship increases in their socialization. Therefore, companionship may be a potential mechanism to explain the relationship between parent-offspring communication and school adaptation. Hypothesis 2 was therefore put forward in this study.

**Hypothesis** **2 (H2).**
*Companionship is a mechanism that mediates between parent-offspring communication and school adaptation.*


### 1.3. Moderating Effect of Safety Perception

Safety is an individual’s sense of anticipation of possible risks and dangers, which Maslow divided into three aspects: emotional safety, interpersonal safety, and self-safety [28]. Researchers of the psychoanalytic school, such as Freud, believed that a person’s sense of safety was closely related to their early parental education and care [29]. If parents give their children enough love during early childhood, the children can develop a sense of safety that allows them to cope with future challenges in life. Maslow also noted that insecure people tend to feel unaccepted and hide low self-esteem behind hostility in their interactions with others, which is not conducive to individuals establishing healthy relationships with others [30]. Healthy parent-offspring communication builds a sense of safety, and research has indicated that healthy communication between parents and children has a significant and positive effect on an individual’s sense of safety [31]. Conversely, if there is a lack of communication between parents and children, this can disrupt the security of the attachment between parents and children, thus increasing the child’s feeling of unsafety.

It was found that the group of leftover children had a significantly lower sense of safety than those who were not left behind, and the longer they were left behind, the stronger the children’s sense of unsafety [32]. Parent-offspring separation led to poor interpersonal safety among left-behind children, who were more prone to exhibiting emotional instabilities. Zotova et al. also showed that children with poorer levels of safety had more strained companionship and were prone to antisocial behavior with their peers [33]. In addition, a sense of safety can regulate individual companionship, and children with higher levels of safety will have more harmonious companionship [34]. Therefore, for left-behind children, a sense of safety may be a moderating factor in the mechanisms of parent-offspring communication and companionship. This study therefore proposes Hypothesis 3.

**Hypothesis** **3 (H3).**
*Safety plays a moderating role in the “parent-offspring communication → companionship*
*” pathway.*


### 1.4. Purpose of the Study

Taking into consideration the current state of school adaptation among left-behind children, this study aimed to answer three questions: (1) whether positive and effective parent-offspring communication can positively predict the level of school adaptation among leftover children in the Chinese diaspora; (2) whether companionship play a mediating role in the relationship between parent-offspring communication and school adaptation; and (3) whether a sense of safety can moderate the effect of parent-offspring communication on companionship among left-behind children.

## 2. Materials and Methods

### 2.1. Participants

In order to reduce the negative bias of the target group of leftover children in overseas countries, two surveys were conducted on all students in June and December of 2022. A convenience sampling method was used to randomly distribute 957 questionnaires to school students in grades three to eight in six schools in Wenzhou, Zhejiang Province. The students were randomly selected within the classrooms. 943 valid questionnaires were returned (98.54% effective rate). Among them, 561 were leftover children in overseas Chinese villages, the final population of this study. The study sample consisted of 292 boys (52.05%) and 269 girls (47.95%); the age range of the subjects was 9–15 years old, with an average age of 12.83 years (SD = 0.95 years). For 197 children, their fathers were away (35.12%); for 158 children, their mothers were away (28.16%); and for 206 children, both parents were away (36.72%).

### 2.2. Measure

#### 2.2.1. Parent-Offspring Communication Scale

The Parent-Offspring Communication Scale compiled by Barnes and Olson and developed by Su et al. was used to evaluate the status of the parent-offspring communication between students and their parents [35,36]. There were 20 questions divided into two dimensions in the scale: openness to communication and problems with communication, covering five major areas: academic problems, daily life, safety awareness, interpersonal communication, and emotions and feelings. A 5-point scale was used, where 1 was considered “strongly disagree” and 5 was considered “strongly agree”. The dimension of communication problems was scored in reverse, and better parent-offspring communication was indicated by a higher score. In this study, there was an internal consistency coefficient of 0.83 for the parent-offspring communication scale, and the validity indexes of the scale were well-fitted (X^2^/df = 3.57, CFI = 0.99, IFI = 0.90, GFI = 0.92, TLI = 0.97, RMSEA = 0.06).

#### 2.2.2. School Adaptation Scale

Students’ school adaptation was measured using the adaptation subscale of the Adolescent Psychological Scale developed by Zhang et al. [37]. There were thirteen items in the scale, divided into three dimensions, namely, academic adaptation, behavioral adaptation, and emotional adaptation. A 5-point scale was used, where 1 was considered “not at all” and 5 was considered “fully”. Furthermore, there was a high internal consistency coefficient of 0.87 for the school adaptation scale, and the validity indicators of the scale were well-fitted (X^2^/df = 2.16, CFI = 0.93, IFI = 0.96, GFI = 0.98, TLI = 0.95, RMSEA = 0.03).

#### 2.2.3. Student Companionship Scale

Students’ relations with their peers were evaluated using the Student Companionship Scale developed by Rose and Asher [38]. There were sixteen items in the scale, divided into 3 categories, namely, welcoming, exclusionary, and loneliness. Using a 5-point scale where 1 represents “not at all” and 5 represents “fully”. The student’s scores were added up from 16 reverse scores after the 16 items were converted into reverse scores, and a higher score meant better companionship. Furthermore, there was an internal consistency coefficient of 0.72 for the student companionship scale, and the validity index of the scale was well-fitted. (X^2^/df = 3.15, CFI = 0.94, IFI = 0.99, GFI = 0.92, TLI = 0.91, RMSEA = 0.08).

#### 2.2.4. The Sense of Safety Scale

Students’ sense of safety was assessed using the Sense of Safety Scale developed by Liao et al. [39]. There were 16 items in the scale, divided into 2 factors. The interpersonal safety factor reflected how secure the individual felt when interacting with others, such as “I feel afraid to establish and keep close relationships with others”; the sense of certainty and control mainly reflected how well the individual felt able to foresee and control their life, such as “I am always worried that my life will become a mess”. A 5-point scale was used, where 1 indicated “not at all” and 5 indicated “fully”. Moreover, there was a high internal consistency coefficient of 0.89 for the safety scale, and the validity indicators of the scale were well-fitted (X^2^/df = 1.79, CFI = 0.95, IFI = 0.95, GFI = 0.97, TLI = 0.90, RMSEA = 0.02).

### 2.3. Research Procedures

Prior consent for this study was obtained from the school administrators and the parents of students, and parents who were abroad signed an online informed consent form. In each class, 2 main testers were assigned to administer the questionnaires. Among them, the Parent-Offspring Communication Scale surveyed the communication between students and their parents; teacher assistance was provided in each class during administration and lasted approximately 20 min. As soon as the questionnaires were returned, SPSS 26.0 was used to perform common method bias tests, descriptive statistics, and correlation analyses. Furthermore, PROCESS for SPSS, developed by Hayes (2013), was used to analyse the mediation effects [40].

## 3. Results

### 3.1. Common Method Deviation Test

An unrotated principal component factor analysis was performed on all variables using Harman’s one-way test. In this study, 24 factors with eigenvalues greater than 1 were found, and the first factor explained 23.17% of the variance, which was below the 40% critical criterion, confirming that there were no serious issues with common method bias in this study.

### 3.2. Descriptive Statistics and Latent Variable Correlation Analysis

The four factors of parent-offspring communication, companionship, school adaptation, and a sense of safety were correlated, and all four showed significant positive correlations (*p* < 0.01). The details are as follows (Table 1): parent-offspring communication and companionship were significantly correlated (*r* = 0.63, *p* < 0.01), parent-offspring communication and school adaptation were significantly correlated (*r* = 0.50, *p* < 0.05), parent-offspring communication and a sense of safety were significantly correlated (*r* = 0.49, *p* < 0.01), school adaptation and companionship were significantly correlated (*r* = 0.64, *p* < 0.01), companionship and a sense of safety were significantly correlated (*r* = 0.40, *p* < 0.01), and school adaptation and a sense of safety were significantly correlated (*r* = 0.24, *p* < 0.001).

### 3.3. The Role of Parent-Offspring Communication on School Adaptation among Leftover Children in the Chinese Diaspora: A Moderately Mediating Effect 

First, companionship was examined as a mediating variable to explore how parent-offspring communication and school adaptation were related (Table 2). Model 4 in the PROCESS program was used for mediating effects analyses. Tests for confidence interval (CI) estimates were conducted using the bias-corrected percentile bootstrap method, with a replicate sampling of 5000, and 95% confidence intervals were calculated.

The results show that school adaptation was directly predicted by parent-offspring communication (β = 0.252, *t* = 5.195, *p* < 0.001); parent-offspring communication positively predicted companionship (β = 0.467, *t* = 7.311, *p* < 0.001); and companionship positively predicted school adaptation (β = 0.129, *t* = 2.520, *p* < 0.05). Moreover, the bootstrap 95% confidence interval excludes zero in its upper and lower limits, indicating that companionship played a partially mediating role between parent-offspring communication and school adaptation (Figure 1), with a partially mediating effect of 0.06, contributing to 25.2% of the overall effect.

Second, the moderating effect was tested using model seven in Hayes’ (2013) macro, and the results are presented in Table 3. When safety was included in the model, parent-offspring communication and safety impacted companionship significantly (β = −0.152, *p* < 0.05), showing that safety can moderate how parent-offspring communication affects companionship.

Additionally, simple slope analysis (Figure 2) revealed that parent-offspring communication significantly predicted companionship for leftover children in the Chinese diaspora with low levels of safety (M − 1SD), simple slope = 0.191, *t* = 2.29, *p* < 0.05; while parent-offspring communication also significantly and positively predicted companionship for left-behind children with high levels of safety (M + 1SD), simple slope = 0.106, *t* = 4.464, *p* < 0.01, showing that as the level of safety of left-behind children increased, the positive predictive effect of parent-offspring communication on companionship also increased.

## 4. Discussion

### 4.1. The Predictive Role of Parent-Offspring Communication on School Adaptation

This study used structural equation modeling to examine the effect mechanism between parent-offspring communication and the school adaptation of leftover children in overseas villages. The direct effects model showed that the school adaptation level of leftover children in the Chinese diaspora was significantly and positively predicted by parent-offspring communication, and that the school adaptation level of children improved as the parent-offspring communication score increased. The results of this study verified Hypothesis 1, which was in agreement with previous research and provided empirical support for further initiatives to improve parent-offspring communication among left-behind children [41]. For leftover children, the absence of parents leads to a lack of daily family interactions, a decrease in the effectiveness of communication, a “disconnection and weakening” of parent-offspring relations, and a tendency to alienate parent-offspring communication [42]. In addition, the content of communication is also mostly focused on academic achievement, ignoring children’s social adaptation abilities [43,44], and therefore left-behind children lack a referential basis for coping with challenges in the school environment and tend to have poor school adaptation. A cross-sectional study by Lu et al. [45] pointed out that leftover children’s psychological adaptation ability is weaker due to the lack of effective parent-offspring communication, and parent-offspring communication can influence leftover children’s school adaptation by acting as a mediating variable [44]. Therefore, parent-offspring communication can improve an individual’s school adaptation level, and effective parent-offspring communication is a protective factor for the school adaptation of leftover children in the Chinese diaspora.

### 4.2. The Mediating Role of Companionship

The results of the mediation model indicated that companionship mediated between parent-offspring communication and the school adaptation of leftover children in the Chinese diaspora, and this result verified Hypothesis 2: that Companionship is a mechanism that mediates between parent-offspring communication and school adaptation. This was consistent with previous findings that family factors (e.g., parent-offspring attachment and parent-offspring communication) can influence individual adaptive behavior by acting on peer factors [45].

First, the companionship of left-behind children who lack effective parent-offspring communication needs to be improved, possibly because children who lack effective parent-offspring communication have a poorer sense of self-identity in the school environment [46], which is accompanied by higher levels of hostility when dealing with others and therefore creates more strained companionship. Second, companionship can affect an individual’s level of school adaptation. For left-behind children with unfavorable parent-offspring communication, peers can provide social support to compensate for the lack of family support during children’s socialization and adaptation [47]. In addition, mutual help and support among peers can help overseas leftover children improve their school maladaptation. In our previous research, we found that companionship plays a mediating role between parent-offspring communication and the school adaptation of leftover children [48]. The current study further demonstrated that companionship partially mediates the impact of parent-offspring communication on the school adaptation of left-behind children.

### 4.3. Moderating Effect of Safety Perception

This study revealed that the first half of “parent-offspring communication → companionship → school adaptation” was moderated by the children’s sense of safety, and the influence of parent-offspring communication on companionship was enhanced when the level of safety of leftover children in overseas countries was low. Moreover, their sense of safety was largely influenced by factors such as family environment and parenting style [49]. On the one hand, in this role of mediating the effects of parent-offspring communication on the school adaptation of left-behind children, parent-offspring communication was more predictive of companionship among individuals with low levels of safety. In other words, higher levels of safety enhanced the influence of the family environment on children’s peer interactions. Some studies have found that feelings of safety can buffer the influence of a negative family environment on the social adaptation of children [50]. On the other hand, according to the psychosocial model of differential interaction theory, a sense of safety affects individuals’ peer interactions, and the formation of companionship is positively influenced by healthy parent-offspring communication among individuals with higher levels of safety.

Therefore, it is important to enhance the sense of security in children who have been left behind. Studies have shown that left-behind children’s sense of security and life satisfaction are positively correlated, and that positive social support plays a mediating role. Thus, we should provide more social support to left-behind children [51]. In addition, the theory of mind elaborates upon the link between caregivers’ intentions and children’s sense of security [52]; healthy care levels also play a role in improving the sense of security of left-behind children.

### 4.4. Limitations

This study suggests new ideas for research on the relationship between parent-offspring communication and school adaptation among leftover children across countries, but there are still some shortcomings. For one, this study was based on students’ self-reports and statements, and multiple methods of data collection could be used to increase objectivity in the future. Furthermore, we did not examine how long these children had been left behind and whether these phenomena were passed on intergenerationally, and will continue to investigate and compare differences.

## 5. Conclusions

The following conclusions were drawn from this study: (1) All four variables—parent-offspring communication, school adaptation, companionship, and a sense of safety—showed positive correlations with each other. (2) Having controlled for the effects of sex and grade level, the relationship between parent-offspring communication and the school adaptation of left-behind children was partially mediated by companionship. (3) A sense of safety played a moderating role between parent-offspring communication and companionship, and the role of parent-offspring communication on companionship was enhanced under a low sense of safety. These conclusions reveal how companionship and a sense of safety affect left-behind children’s school adaptation, which provides empirical support for improving the school adaptation of leftover children.

## Figures and Tables

**Figure 1 behavsci-13-00557-f001:**
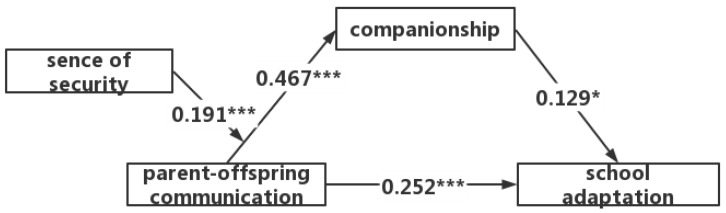
Mediating effects of companionship (* *p* < 0.05; *** *p* < 0.001).

**Figure 2 behavsci-13-00557-f002:**
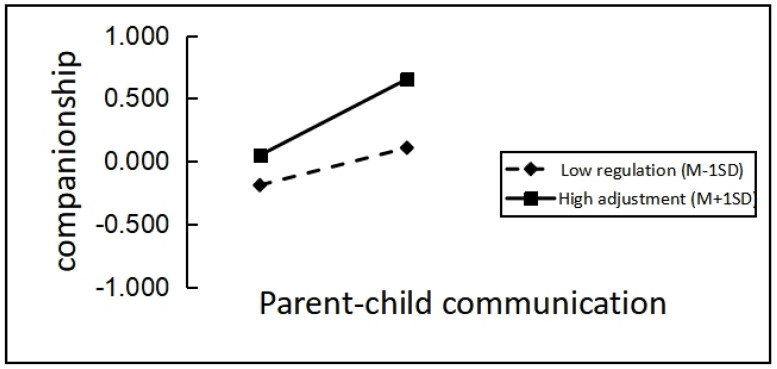
The moderating role of safety in parent-offspring communication and companionship.

**Table 1 behavsci-13-00557-t001:** Descriptive statistics and correlation analysis among the variables for left-behind children (*n* = 561).

	M	SD	1	2	3	4
1. Parent-offspring communication	3.30	0.68	1			
2. Companionship	3.88	0.39	0.63 **	1		
3. School adaptation	3.24	0.78	0.50 *	0.64 **	1	
4. Sense of safety	2.75	0.58	0.49 **	0.40 **	0.24 ***	1

Notes: * *p* < 0.05; ** *p* < 0.01; *** *p* < 0.001.

**Table 2 behavsci-13-00557-t002:** Direct and mediated effects at different levels of perceptions of safety.

	Safety	Effect Value	Boot Standard Error	Boot CI Lower Limit	Boot CI Higher Limit
Direct role	M − 1SD	0.161	0.030	0.022	0.138
	M	0.252	0.024	0.017	0.110
	M + 1SD	0.284	0.020	0.012	0.090
The mediating role of companionship	M − 1SD	0.076	0.083	0.425	0.751
	M	0.060	0.064	0.341	0.593
	M + 1SD	0.045	0.080	0.188	0.504

Notes: 95% CI does not contain 0. Significant mediating effect.

**Table 3 behavsci-13-00557-t003:** Tests for mediating effects with moderation.

Regression Equation (*n* = 516)		Overall Fit Coefficient				
Result Variables	Predictive Variables	R	R-sq	F	β	t
Companionship		0.650	0.423	50.357		
	Grade				0.026	1.649
	Gender				0.054	1.586
	Parent-offspring communication				0.467	7.311 ***
	Safety				0.191	3.026 ***
	Parent-offspring communication x sense of safety				−0.152	−2.397 **
School Adaptation		0.528	0.278	39.914		
	Grade				0.057	3.620
	Gender				0.112	3.253
	Parent-offspring communication				0.252	5.195 ***
	Companionship				0.129	2.520 *

Notes: * *p* < 0.05; ** *p* < 0.01; *** *p* < 0.001.

## Data Availability

The datasets generated during and/or analysed during the current study are available from the corresponding author on reasonable request.

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
