# Peer review of "The Relationship between Parent-Offspring Communication and the School Adaptation of Leftover Children in Overseas Countries: The Mediating Role of Companionship and the Moderating Role of a Sense of Safety"

_behavsci, 2023, doi:10.3390/bs13070557_

Round 1
Reviewer 1 Report
article on The Relation between Parent-Offspring Communication and the School
Adaptation of Leftover Children in Overseas Countries: The Mediating Role of
Companionship and the Moderating Role of a Sense of Safety
The topics chosen are quite interesting and up to date.
the abstract is prepared in accordance with scientific principles, which consists of background, method, results, conclusions and keywords, suggestions: for an abstract, please explain briefly about the research design, samples, variables, instruments and data analysis at the point of the method
an introductory chapter that contains background, please emphasize what phenomena you encountered so that it becomes your research topic, of course it is supported by relevant data
the research method is good, what about the inclusion and exclusion criteria along with how you select the sample please explain briefly
the results and discussion are quite good, add your opinion to the discussion which is supported by theory
but in the attachment of your results, especially in the table, there is still a table that is cut off in 1 page, please fix it first
conclusions must answer the research objectives
references are used accordingly
Author Response
Dear Reviewer,
Thank you very much for reviewing our manuscript and your valuable suggestions, which are very important in improving the quality of the article. The following are responses.
1.for an abstract, please explain briefly about the research design, samples, variables, instruments and data analysis at the point of the method——We have revised the presentation of the method in the abstract to include study design, sample selection, data analysis, etc. (lines 14-19), and attached the revised content:
We administered questionnaires to 957 children from six schools in June and December in 2022, all students in the sample were randomly selected within the classrooms. 561 (47.95% female, mean age = 12.84, SD = 0.95) of them were leftover children in total. Self-report questionnaires on parent-offspring communication with thier parents, school adaptation, companionship, and feelings of safety were used in this investigation, and then we used SPSS software and the PROCESS plugin to analyze the relationships between variables.
2.please emphasize what phenomena you encountered so that it becomes your research topic, of course it is supported by relevant data。——We added our team's previous findings in fieldwork that there was a significant difference between the level of school adaptation of left-behind children and non-left-behind children, we added relevant statistical data (lines 49-52),attached the revised content:
In addition, in our fieldwork, we also found that left-behind children in overseas Chinese have a poorer level of school adaptation compared to non-left-behind children,The differences were significant (t = −5.64, Cohen’s d = −0.73, p < 0.001) (Zhang et al.,2022)
3.what about the inclusion and exclusion criteria along with how you select the sample please explain briefly——We added a description of the method of selecting samples (lines 15, 160-162),attached the revised content:
A convenience sampling method was used to randomly distribute 957 questionnaires to school students in grades 3 to 8 in six schools in Wenzhou, Zhejiang Province, the students were randomly selected within the classrooms.
4.there is still a table that is cut off in 1 page, please fix it first——We have fixed the tables, all in the same page.
5.conclusions must answer the research objectives——We have revised the conclusion section to add the content proving the subject of the study (lines 369-372) ,attached the revised content:
These conclusions reveal how companionship and a sense of safety affect left-behind children's school adaptation, which provide empirical support for improving the school adaptation of leftover children.
Reviewer 2 Report
This is a well written manuscript and is tightly structured around a highly quantitative survey study.
The parent-offspring communication scale is difficult to track down and a copy of it would be useful as a supplement or appendix. Additionally, it is not clear who the children answer the scale about - both parents, the carers, what role does the non-absent parent play?
Given the specific location and culture of the children studied, is there good reason to be able to generalise from this cohort to others, or would a different cultural and ethnic mix provide different thinking? Some history of the cohort would be of value in understanding the relevance of this group to others - have families in this region been working overseas for generations or is this a newer phenomenon to this group?
Author Response
Dear Reviewer,
Thank you very much for reviewing our manuscript and your valuable suggestions, which are very important in improving the quality of the article and gave us some inspiration to the follow-up research, we will continue to improve . The following are responses.
1.The parent-offspring communication scale is difficult to track down and a copy of it would be useful as a supplement or appendix——At present, there are many new developments in the research on parent-child communication, and the relevant scales are also being improved and updated, we are also continuing to pay attention. If readers need, they can send e-mail to the corresponding authors, we will provide the corresponding scale.
2.it is not clear who the children answer the scale about - both parents, the carers——in the parent-child communication scale, the child answered the communication with parents, and the assistances had told the student during the study. This is further explained in the abstract of the article and in the parent-child communication research tool (line 173, 217-218),attached the revised content:
Among them, the Parent-Offspring Communication Scale surveyed the communication between students and their parents.
3.is there good reason to be able to generalise from this cohort to others, or would a different cultural and ethnic mix provide different thinking?——Previous studies have found that left-behind children are a vulnerable group and show poor levels of academic performance and interpersonal relationships. Our study focuses on neglected transnational left-behind children and wants to give more care to these children, who are also part of the left-behind children. In addition, in follow-up research, we will also conduct research on transnational left-behind children from different cultural and ethnic to compare whether there are differences. Thank you for your comments, and it also expands our follow-up research ideas.
4.have families in this region been working overseas for generations or is this a newer phenomenon to this group?——Thank you for your valuable suggestion, this is a limitation of this study and where further improvements can be made, we will reach out to local authorities and education departments and interview these children to find out whether the parents of this group have been abroad for generations or are just new phenomena. After learning the relevant information, we will try to compare whether there are differences between the first generation of overseas Chinese, the second generation of overseas Chinese, and the third generation of overseas Chinese, attached the revised content(line 358-360):
Besides, we did not examine how long these children had been left behind and whether these phenomena were passed on intergenerationally, and will continue to investigate and compare differences.
Thank you again for your valuable suggestions!
Reviewer 3 Report
Dear Authors,
This is an interesting study examining with moderated mediation analysis the conditional indirect effect of the Sense of Safety on the relationship between Parent-Offspring Communication and School Adaptation via Student Companionship. Few recommendations:
1. The abstract does not need to include limitations (lines 19-21).
2. Did you examine how long these children had been left behind? If no, please include it in the limitations section.
3. In the instruments section, is it Parent-Offspring Communication Questionnaire or Scale?
Similarly, the Sense of Safety is it a Scale or a Questionnaire?
4. In table 1 you merged results from descriptive statistics with correlation analysis. Either you split the table or you can change the title.
5. Table 3 presents results from regression analysis. This should be included in the title of the table.
6. In discussion section Line 281: ‘This study constructed structural equation modeling…’ This is not apparent from the results section. Instead, you seem to have employed regression analysis. Where the assumptions of multiple regression analysis satisfied?
7. Figure 1 is not how a Moderated Mediation Analysis model 7 should be presented. It is more like a model 4 without the conditional effects on path a. Please revise. Or, you can keep figure 1 as model 4 and delete the moderator and create another figure of model 7 with the moderator Sense of Safety and include the conditional effects on path a.
8. The “PROCESS" macro, model 7 provides test for the interaction parent-offspring communication X Sense of Safety. You did not need to construct regression analysis to prove the interaction.
9. The reference 44 that you include in line 323 does not seem to correspond to the information given in this sentence: ‘A study of leftover children found that parent-offspring communication from the home environment predicted school adaptation’. Instead, in this paragraph you can include information based on the recent studies from your team (doi.org/10.3390/ijerph19126989 and 10.3389/fpsyg.2022.1041805) where you have already investigated the mediating effect of peer relationships.
10. Please include recommendations section. How can we enhance the Sense of Security of Children Left Behind?
11. Please avoid repetition of the same phrase in one paragraph: ‘leftover children in the diaspora’ (lines 132, 137, 149 and 270, 274 and 284, 289, 307,312, 325, 333, 354}. Choose another phrase.
Moderate editing of English language is required.
Author Response
Dear Reviewer,
Thank you very much for reviewing our manuscript and your valuable suggestions, which are very important in improving the quality of the article and gave us some inspiration to the follow-up research, we will continue to improve . The following are responses.
1.The abstract does not need to include limitations (lines 19-21)——We've removed the limitations in the abstract.
2.Did you examine how long these children had been left behind? If no, please include it in the limitations section——Thank you for your valuable suggestions, this is a limitation of this study and where further improvements can be made, we will contact local authorities and education departments and interview these children to find out how long these children are left behind across borders. Once we have this information, we will try to compare whether there are differences between children left behind for different periods of time and refine the study. We have includedit in the limitations section(line 358-360) , attached the revised content:
Besides, we did not examine how long these children had been left behind and whether these phenomena were passed on intergenerationally, and will continue to investigate and compare differences.
3.is it Parent-Offspring Communication Questionnaire or Scale?Similarly, the Sense of Safety is it a Scale or a Questionnaire?——We have revised, both they are scales (line 170,202).
4.In table 1 you merged results from descriptive statistics with correlation analysis. Either you split the table or you can change the title——we have changed the title as “Descriptive statistics and Correlation analysis among the variables for left-behind children”(line 241).
5.Answers to recommendations 5、6 and 8, they are about regression analysis——Thanks for your suggestions on the statistical results, we have removed the regression analysis.
6.Figure 1 is not how a Moderated Mediation Analysis model 7 should be presented. It is more like a model 4 without the conditional effects on path a. Please revise——we have revised the figure 1.
7.The reference 44 that you include in line 323 does not seem to correspond to the information given in this sentence, Instead, in this paragraph you can include information based on the recent studies from your team where you have already investigated the mediating effect of peer relationships. ——Thanks for your valuable suggestions, we have revised it and include information based on the studies(line 323-326),attached the revised content:
In our previous research, we have found that companionship plays a mediating role between parent-offspring communication and the school adaptation among leftover children(Zhang et al.,2022).
8.Please include recommendations section. How can we enhance the Sense of Security of Children Left Behind?——Thank you for your comments. After consulting relevant literature and combining our research findings, we have supplemented the content of this part(line 346-352).attached the revised content:
Therefore, it is important to enhance the sense of security of children left behind. Studies have shown that left-behind children's sense of security and life satisfaction are positively correlated, and positive social support plays a mediating role, thus we can give more social support to left-behind children (Liu et al., 2023). In addition, the theory of mind elaborates the link between caregivers' intentions and children's sense of security (Yang et al., 2022), good care levels also play a role in improving the sense of security of left-behind children.
9.Please avoid repetition of the same phrase in one paragraph: ‘leftover children in the diaspora’——We have replaced it as “left-behind children” in the appropriate places.
10.Moderate editing of English language is required——We are sorry for the language problem, and we had improved the language using a language editing service.
Thank you again for your valuable suggestions !
